# The role of parasite-produced dopamine in *Toxoplasma gondii*-altered host behaviour

**Beatriz Calvo-Urbano** [1], **Maya Kaushik**[2], **An-Chi Cheng** [1], **Greg C. Bristow** [3,4], **Emese Prandovszky** [3,5], **Martin Walker** [1], **Poppy H. L. Lamberton** [2,6], **Glenn A. McConkey** [3] & **Joanne P. Webster** [1] ✉

Certain parasites can manipulate host behaviour for their own benefit, but the mechanisms remain largely unknown. *Toxoplasma gondii*, the agent of the toxoplasmosis, is a canonical example, altering behaviour in rodents and other hosts, including humans. Dopamine dysregulation has been suggested as a mechanism, with parasite-encoded tyrosine hydroxylases (*Tg*TH) proposed as a direct source of dopamine, though their role is debated. Here, using *Rattus norvegicus* as a model, with subtle and specific behavioural and biostatistical assays and analyses, we examine the contribution of *Tg*TH to behavioural change. Two engineered *T. gondii* Prugniaud lines with moderate and high *Tg*TH overexpression (OE) are compared to wild-type and recombinant wild-type parasites, alongside uninfected controls. All genetically modified lines induce weaker behavioural changes than true wild-type, but changes correlate with *Tg*TH expression levels. Our findings provide empirical support that *Tg*TH contributes to *T. gondii*-associated behavioural alterations, highlighting both theoretical significance and applied implications.

The ability of parasites to alter host cognition and behaviour captivates the interest of both the scientific and lay communities, partly because it raises questions about longstanding philosophical issues such as the existence of free will. Certain parasites, including a notable minority within vertebrates, appear able to manipulate their host's central nervous system. Whilst many behavioural changes evoked are general, such as malaise, others appear remarkable in their specificity, subtly altering only a limited repertoire of key host behavioural traits for apparent fitness benefits to the parasites. The mechanisms behind such behavioural alterations in vertebrate hosts remain elusive, despite their profound implications and applications, particularly for those manipulatory parasites with a broad host range, able to infect species both relevant (parasite manipulation) and not (parasite constraint) to their own transmission potential.

*Toxoplasma gondii* is the causative agent of toxoplasmosis, a globally important protozoan parasitic disease, where recent meta-analyses report an average worldwide seroprevalence in humans of 25.7%, ranging between 0.5 and 87.7%[1]. *T. gondii* is also a canonical example for parasite manipulation of host behaviour. It has an indirect lifecycle and can infect all warm-blooded animals as intermediate or secondary hosts, with only members of the Felidae as definitive hosts. Since sexual reproduction of this parasite can be accomplished only in felines, there are likely strong selective pressures on *T. gondii* to evolve mechanisms to enhance transmission from the intermediate host to the definitive feline host. The predilection of *T. gondii* for the host's central nervous system also places this parasite in a privileged position to alter host behaviour. Accordingly, an extensive body of research has demonstrated that *T. gondii* appears to manipulate the behaviour of its latently-infected rodent intermediate host to facilitate transmission to

[1]Department of Pathobiology and Population Sciences, Royal Veterinary College, University of London, London, UK. [2]Department of Infectious Disease Epidemiology, School of Public Health, Imperial College Faculty of Medicine, London, UK. [3]School of Biology, Faculty of Biological Sciences, University of Leeds, Leeds, UK. [4]Present address: School of Pharmacy and Medical Sciences, University of Bradford, Bradford, UK. [5]Present address: Stanley Division of Developmental Neurovirology, Department of Paediatrics, Johns Hopkins School of Medicine, Baltimore, MD, USA. [6]Present address: School of Biodiversity, One Health and Veterinary Medicine, University of Glasgow, Glasgow, UK. ✉e-mail: jowebster@rvc.ac.uk

**Table 1 | Lines (strains) of Prugniaud type II (Pru) *Toxoplasma gondii* used**

| Label | Prugniaud line | *TgAaaH2* modified (recombinant) | Over-expressed (OE) | Range of *TH* expression |
|---|---|---|---|---|
| WT | Wildtype Pru | No | No | Normal |
| Tg*TH*0 | Delta H ku80 Pru | No | No | Normal |
| Tg*TH*-MID | Ox11 Pru | Yes | Yes | Middle |
| Tg*TH*-HIGH | Ox6 Pru | Yes | Yes | High |

its feline host[2,3], with one of the most widely cited and replicated examples being that *T. gondii* appears to override rodents' innate fear of cat predators, and the smell of cat urine in particular, turning this instead into an apparent "Fatal Feline Attraction"[4–9].

Comparable behavioural changes, from the subtle to severe, have been reported across numerous secondary hosts, including latently infected humans[10–16], likely as an unavoidable consequence of parasite-altered behaviour. Notably, a convincing body of epidemiological and neuropathological studies indicate that some cases of human neuropsychiatric disorders, particularly that of schizophrenia, are associated with *T. gondii*[15,17–23]. Whilst the mechanisms behind such behavioural alterations of *T. gondii*-infected rats and humans are unknown, repeated findings since the 1980s support potential neuromodulatory roles, either directly or indirectly, in particular that of dopamine dysregulation[15,20,22–30]. For instance, raised or disrupted dopamine levels have been reported in both rodent and human *T. gondii* infections and within human patients with schizophrenia[18,20,22,24,26]. Dopamine receptor antagonists commonly used in the treatment of patients with schizophrenia, such as haloperidol, have been shown both to have anti-*T. gondii* tachyzoite activity in vitro[27], and prevent the development of such behavioural changes in *T. gondii*-infected rats in vivo[20,28]. Furthermore, research has indicated that the parasite itself may be a source of this dopamine[25,30,31]. In mammals, dopamine is synthesised in two steps: from the precursor amino acid, tyrosine, to 3,4-dihydroxy-l-phenylalanine (L-DOPA) by tyrosine hydroxylase metabolism, followed by production of dopamine by aromatic L-amino acid decarboxylase. *T. gondii* was found to encode a protein with high homology and showing similar catalytic properties to the tyrosine hydroxylases found in mammals – the *Tg*TH ortholog synthesises tyrosine to L-DOPA and is encoded by *T. gondii* aromatic amino acid hydroxylase genes *TgAaaH1* (constitutively expressed) and *TgAaaH2* (increased in the bradyzoite stage)[31]. High concentrations of both *Tg*TH and dopamine have been detected in *T. gondii* cysts in rodent's brains, with infected dopaminergic cells releasing 350% more dopamine than uninfected cells[25]. Furthermore, a laboratory mouse challenge study found that recombinant *Tg*TH resulted in increased dopamine levels in vivo in a dose-dependent manner[32].

However, the role of *T. gondii*-produced *Tg*TH in the dopamine-dependent pathway and behavioural alterations remains controversial. One experiment found the expression level of *TgAaaH2* did not affect overall brain dopamine contents of female laboratory mice in vivo, nor did the dopamine contents released by cultured dopaminergic cells (PC12) differ depending on infection with wildtype, *TgAaaH2*-knockout, or *TgAaaH2*-overexpressing lines in vitro[33]. Moreover, two further laboratory mouse studies found no apparent association between *Tg*TH expression level and behavioural change[34,35]. Whilst dopamine concentration relative to dopamine turnover could explain the former[36], the latter findings may be potentially explicable, at least in part, by the choice of host and experimental assays utilized[3]. Therefore, here, using the biologically and clinically-appropriate rat model of both sexes, with highly subtle and specific behavioural assays such as that of the activity, velocity and "Fatal Feline Attraction" four-choice odour test[3,4,7,8,37], we aimed to elucidate a causal link between parasite-produced *Tg*TH and *T. gondii*-induced behavioural change. This study was uniquely designed to precisely assess whether behaviour

alterations are influenced by the level of *Tg*TH expression via incorporating two genetically-modified overexpressing (OE) Prugniaud (Pru) *T. gondii* lines: *Tg*TH-MID (mid-*Tg*TH2 over-expression) and *Tg*TH-HIGH (high-*Tg*TH2 over-expression), as well as two PRU wildtype lines: "true wildtype" (WT) and "recombinant wildtype" (*Tg*TH0) (the latter of which being a Pru transgenic parasite line not affecting *Tg*TH levels and thus served as an additional recombinant control in terms of evaluating any subtle impact of genetic modification in general, independent of *Tg*TH overexpression) (see Table 1); as well as uninfected negative (sham exposed) controls.

Our results found clear behavioural differences between infected and uninfected rats, and where female rats were consistently more active and exploratory than their matched male counterparts. Whilst all three recombinant parasite lines, both over-expressers and control, induced smaller intensities of behavioural alterations relative to true wild-type *T. gondii*, potentially indicative of lowered viability of recombinant lines in general[38,39], the intensity of behavioural changes were consistently dose-dependent on the levels of *Tg*TH over-expression. Our findings thereby provide support for the hypothesis of, at least in part, a mechanistic role of *Tg*TH in *T. gondii*-associated behavioural changes.

## Results

We constructed recombinant *Tg*TH lines of avirulent *T. gondii* PRU overexpressing contrasting levels of *TgAaaH2* and examined their impact on host behaviour using the biologically and clinically appropriate rat model of parasite manipulation with both sexes, and the highly subtle and specific behavioural four-choice odour test, otherwise known as the "Fatal Feline Attraction" assay[3,4,7,8,37]. This non-invasive assay, where free-ranging rats have the option to spend time across either neutral or contrasting urine-odour areas, follows from the original finding that *T. gondii*-infected rats demonstrate not simply a reduction in their innate avoidance of areas with evidence of feline definitive host (only) odour, but instead increase their time in these zones, in contrast to their uninfected counterparts[4]. Using a Bayesian statistical approach to evaluate the predictive power of models including line and sex singularly, as additive or as interactive covariates, we found clear evidence for parasite-altered behaviour between infected and uninfected rats, with a subtle, but dose-dependent, increase in behaviour change with levels of *Tg*TH overexpression.

### Proportion of time spent in the cat zone

Parasite line (i.e., across the four *T. gondii*-infected and one uninfected groups of rats) was the only factor associated with the proportion of time spent in the cat zone. This outcome variable was lowest for the rats in the uninfected Control line and highest for those in the WT line (Fig. 1 and Table S3). The mean differences between the five groups (Table S4) indicated that uninfected Control, *Tg*TH0, *Tg*TH-MID were all different from *Tg*TH-HIGH, and that uninfected Control was different from *Tg*TH-MID, suggesting that there is an increasing *Tg*TH dose-dependent trend from the uninfected Controls to the infected *Tg*TH0, *Tg*TH-MID and *Tg*TH-HIGH.

### Velocity in the cat zone

Line was the only factor associated with the velocity in the cat zone. The velocity (cm/s) in the cat zone was highest in the uninfected

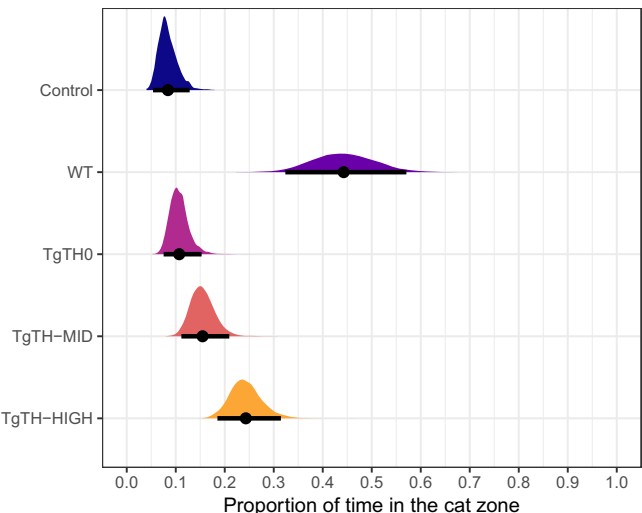

**Fig. 1 | *Tg*TH dose-dependent increase in time spent within the cat zone.** Proportion of time rats spent in the cat zone by infection status across three differentially expressing TgTH recombinant PRU *T. gondii* line groups, one wildtype PRU *T. gondii* line group (purple) and one group of uninfected controls (where: uninfected controls−navy; wildtype−purple; *Tg*TH0 no-overexpression−pink; *Tg*TH-MID middle-level overexpression−red; *Tg*TH-HIGH−high overexpression−orange). Posterior distributions of the mean and 95% credible intervals (CrI), by experimental line. Number of rats per line group: Control = 24, WT = 23, *Tg*TH0 = 12, *Tg*TH-MID = 11, and *Tg*TH-HIGH = 15.

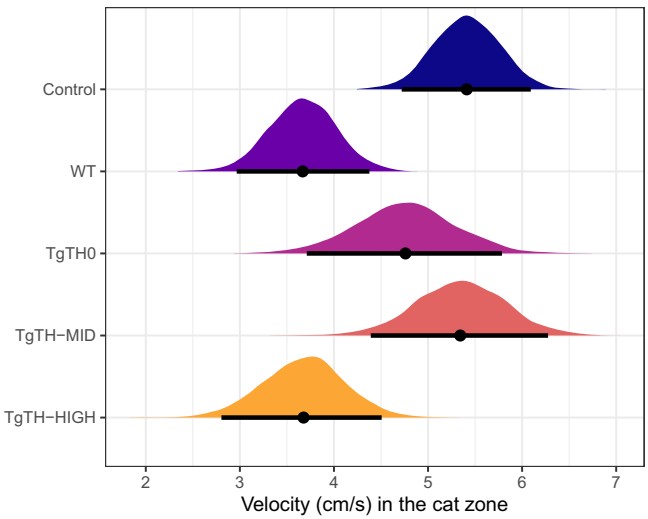

**Fig. 2 | Velocity in the cat zone.** Posterior distributions of the mean and 95% credible intervals, by experimental line (where: uninfected controls−navy; wildtype −purple; *Tg*TH0 no-overexpression−pink; *Tg*TH-MID middle-level overexpression −red; *Tg*TH-HIGH−high overexpression−orange). Number of rats per line group: Control = 24, WT = 23, *Tg*TH0 = 12, *Tg*TH-MID = 11 and *Tg*TH-HIGH = 15.

Control group, lowest in both WT and *Tg*TH-HIGH, and showed intermediate values in *Tg*TH0 and *Tg*TH-MID (Fig. 2 and Table S5). The differences between lines that were relevant (95% credible intervals, CrIs, not including zero) were Control vs. *Tg*TH-HIGH; Control vs. WT; *Tg*TH-MID vs. *Tg*TH-HIGH, and *Tg*TH-MID vs. WT (Table S6).

**Difference in velocity between the cat zone and the non-cat zone**
Line was the only factor associated with difference in velocity in the cat zone and the non-cat zone (velocity in cat zone−velocity in non-cat zone). Control, *Tg*TH0 and *Tg*TH-MID moved faster in the cat zone than in the non-cat zone, with the former showing the greatest

difference (Fig. 3 and Table S7). Both WT and *Tg*TH-HIGH showed no difference (95% CrI includes zero) in velocity between the two zones. Control was different from *Tg*TH-HIGH, and WT was different from all lines except for *Tg*TH-HIGH (Table S8).

**Distance moved in the cat zone**
Line and sex were both associated with the distance moved in the cat zone. Males moved shorter distances in the cat zone than females (Fig. 4 and Table S9). Control and *Tg*TH0 moved shorter distances than *Tg*TH-HIGH and WT. Control males and females were different from other lines of their own sex except for *Tg*TH0. *Tg*TH0 was different from *Tg*TH-HIGH but was not different from *Tg*TH-MID (Table S10).

**Frequency entering the cat zone**
Sex was the main factor associated with frequency of entering the cat zone, with females entering approximately twice as frequently as males (Fig. 5 and Tables S11 and S12).

## Discussion
With increasing pressure to understand causation behind both acute and chronic diseases and disorders, a renewed recognition of the potential role of infectious agents in occurring. *T. gondii*, the causative agent of toxoplasmosis, is a highly successful apicomplexan protozoan capable of infecting all warm-blooded animals worldwide, with substantive evidence of its ability to alter host behaviour. The pathophysiological mechanisms behind such behavioural changes remain, however, unknown, although dysregulation of catecholamines and parasite-produced dopamine has frequently been proposed[15,20,22,24−30]. Here, we empirically demonstrate parasite-altered behaviour, with observable differences between infected and uninfected rats. All recombinant parasite lines induced smaller intensities of behavioural alterations relative to true wild-type *T. gondii*, consistent with studies across a broad range of pathogen and host systems which have found that gene-modifying processes can reduce viability and/or fitness relative to wild-type[38,39]. However, the intensity of behavioural changes observed were dose-dependent to the levels of *Tg*TH expression in the recombinant parasite with which the rats were infected (Figs. 1−5, S1, and S6). Our findings thereby provide unique support for the hypothesis of, at least in part, a mechanistic role for *Tg*TH in *T. gondii*-associated behavioural changes.

Consistent with earlier findings[4,8], there was clear evidence of the apparent reversal of the rats innate aversion to cat urine, where all infected, irrespective of parasite line, spent more time in the cat zone relative to their uninfected control counterparts (Fig. 1). Furthermore, whilst infected rats showed, overall, higher activity levels relative to their uninfected counterparts[40], automated tracking revealed that these infected rats were then slowing down (here shown as a reduced velocity as distinct from any fear-related "freezing"), to explore within this zone (Figs. 2 and 3: although noting that, whilst TgTH-MID were slightly, but not significantly faster than TgTH0 (Fig. 2), the dose-dependent trend of decreasing velocity in the cat zone was apparent in the difference in velocity of individual rats (Fig. 3)). This is consistent with the hypothesis that an increased activity overall enhances the visibility of rats to cats[40], combined with a loss of predatory vigilance and risk perception once within the close proximity to their feline predator[7].

Clear gender effects were also observed, with female rats travelling greater distances (Fig. 4) and entering the cat zone twice as frequently as their male counterparts, independent of infection status (Fig. 5). These findings are consistent with previous analyses, and may be related to both the natural higher activity levels of female rats[40], but also the lower stress levels of females in anxiety-related situations[41].

The "dose-dependency" of intensity of behavioural change found here being in line with overexpression level also provides unique support to the hypothesis that *Tg*TH could be involved in the

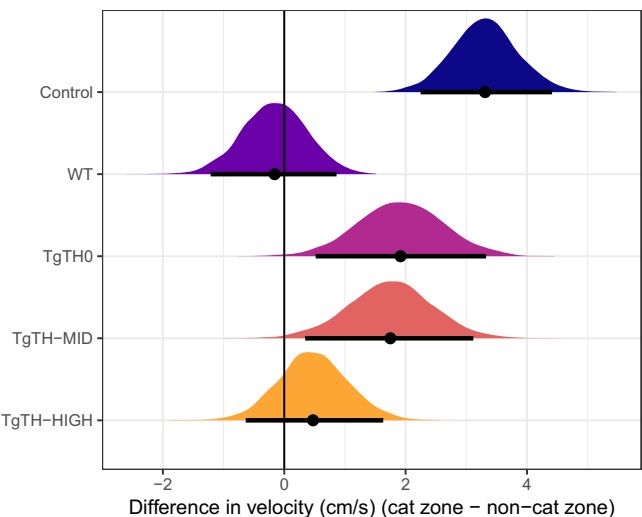

**Fig. 3 | Differences in velocity between zones (velocity in cat zone–velocity in non-cat zone).** Posterior distributions of the mean and 95% credible intervals, by experimental line (where: uninfected controls–navy; wildtype–purple; *Tg*TH0 no-overexpression–pink; *Tg*TH-MID middle-level overexpression–red; *Tg*TH-HIGH–high overexpression–orange). Number of rats per line group: Control = 24, WT = 23, *Tg*TH0 = 12, *Tg*TH-MID = 11 and *Tg*TH-HIGH = 15.

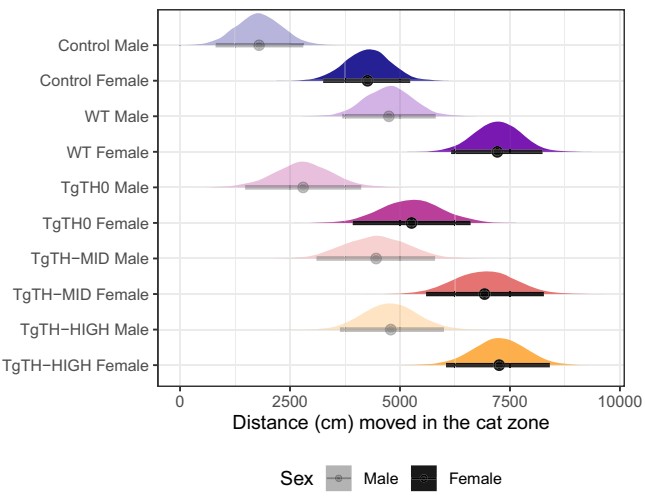

**Fig. 4 | Distance (cm) moved in the cat zone.** Posterior distributions of the mean and 95% credible intervals, by experimental line (where: uninfected controls–navy; wildtype–purple; *Tg*TH0 no-overexpression–pink; *Tg*TH-MID middle-level over-expression–red; *Tg*TH-HIGH–high overexpression–orange). Number of rats per line group: Control = 24, WT = 23, *Tg*TH0 = 12, *Tg*TH-MID = 11 and *Tg*TH-HIGH = 15.

mechanism of behavioural modification, potentially through L-DOPA production and subsequent dopamine elevation and/or metabolism. Indeed, our findings are consistent with Zhang and colleagues, who showed *Tg*TH from both RH and PRU *T. gondii* strains catalysed dopamine in a dose-dependent manner[32]. For instance, whilst all recombinant parasite lines produced lower levels of behavioural change relative to the true wildtype control, the proportion of time spent in the cat zone was sequentially higher than uninfected controls from *Tg*TH0 to -MID to -HIGH lines (Fig. 1). This overexpression dose-dependency from *Tg*TH0 to -MID to -HIGH in behavioural change relative to uninfected controls was again consistently apparent when observing the differences in velocity between zones (Fig. 3) and the distances moved within the cat zones across both genders (Fig. 4).

There are several potential explanations for our outcomes here relative to the two previous laboratory mouse studies, which did not find any apparent association between *Tg*TH and behavioural changes. Afonso and colleagues found that female mice acutely infected with wildtype or AaaH2-knockout lines showed changes in risk behaviour[34], and McFarland and colleagues found, as part of a psychostimulant-induced hyperactivity study of (sex not specified) mice, that *T. gondii* led to a blunted response to amphetamine or cocaine and decreased expression of a dopamine transporter, but that deletion of *AAH2* had no effects on these changes[35]. Whilst such knockout studies are undoubtedly of value, the rationality of deleting a single copy of duplicated genes, leaving the constitutively expressed gene intact to synthesise *Tg*TH, is unclear. As regards to our study, firstly it must be emphasized that rats and mice are not the same, in terms of their susceptibility, behaviour, morbidity and overall response to *T. gondii*. Indeed, mice have been suggested, despite their relative ease of handling and laboratory logistics, as a less-than-ideal model for examining *T. gondii*-associated manipulation behaviour, due to the generalized pathologies induced relative to the more resistant rat model, which is also believed to be more comparable to human infections[3,20,42]. Potential explanations for these differences relate to the higher infection rate of *T. gondii* in the brains of mice than rats during latent toxoplasmosis and the formers' increased potential for severe morbidity during the acute phase of infection. Indeed, whilst the general health and behaviour of laboratory rats usually appear unaffected by infection, laboratory mice often show high parasite-induced mortality, even with avirulent Type II *T. gondii* strains, and/or signs of acute infection such as running in circles with their heads bent to one side[43,44]. It may be relevant to note that all parasite-infected mice in one of the aforementioned studies were reported to develop a sickness phenotype[34], whilst none of the rats in our study did. Indeed, Afonso and colleagues pertinently question whether their findings would be present in mouse models more resistant to parasitic infection or other known key rodent hosts such as rats[34].

The inclusion of both sexes within our study may also play a role, since sex differences in both *T. gondii*-altered behaviour and catecholamine dysregulation are consistently observed in humans[10,11] together with sex-specific changes in gene expression, catecholamine dysregulation and/or behaviour induced by chronic *T. gondii* infection in rodents[45,46]. For instance, Gatkowska and colleagues observed a decrease in noradrenergic system activity in female mice and a slight increase in some brain areas of males during acute toxoplasmosis, with an accompanying rise in serotonin activity predominantly in male mice[46], whilst other studies also observed catecholamine dysregulation only in male rodents, whilst females were unchanged[47].

However, the third key reason for the differences between our findings here with that of the two previous laboratory mouse studies may relate to the experimental design and choice of behavioural assays. *T. gondii*-associated behavioural alterations observed in rats are both specific and subtle, requiring appropriate measures and analytical timeframes[3,4,37]. Since sexual reproduction of *T. gondii* can be accomplished only in felines, there are likely powerful selective pressures on the parasite to evolve mechanisms to enhance transmission from the intermediate host to the definitive feline host. The strong innate aversion of rodents to cat odour provides a potentially substantial obstacle for the parasite against successful predation by the feline definitive host. Our and others previous research demonstrated that *T. gondii* appears to selectively alter a rat's perception of the risk of being preyed upon by cats alone, leaving many other behavioural traits intact[4,5,7,28,48], thereby making the "Fatal Feline Attraction" four-choice odour assay an ideal and specific measure. The duration of our assays may also be important. Whilst the original "Fatal Feline Attraction" assays were ten hours long[4], this and subsequent assays found 2.5 h to be the minimum duration required for reliability[3,28,37]. Afonso and colleagues performed a very nice set of non-invasive behavioural assays on female laboratory mice, but these ranged from 6 min (trapping simulation) to 10 min (open field test)[34]. The McFarland and

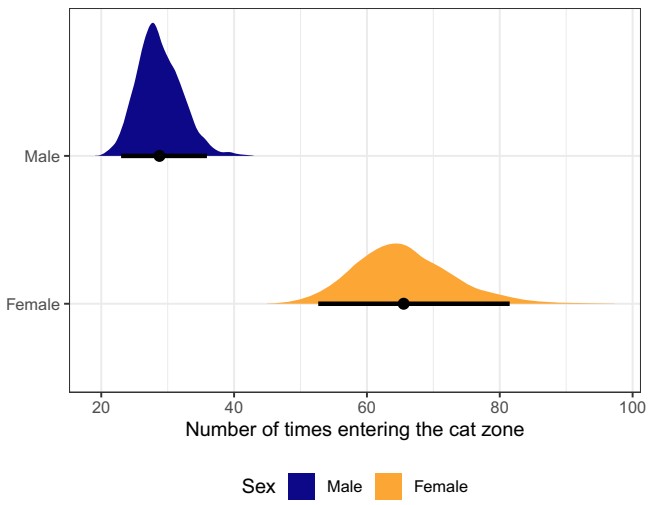

**Fig. 5 | Frequency entering the cat zone.** Posterior distributions of the mean and 95% credible intervals, by sex (where: male navy and female orange). Number of rats per line group: Control = 24, WT = 23, *Tg*TH0 = 12, *Tg*TH-MID = 11 and *Tg*TH-HIGH = 15.

colleague's laboratory mouse studies did include longer duration assays, from the startle chambers at 5 min, to open field up to 30 min or 90 min, but where the mice were also given D-amphetamine. In addition to the aforementioned risk of non-specific morbidities and mortalities of mice relative to rats, these less feline-predation-specific and more stressful behavioural assays, especially with the additional amphetamine procedures, may mask any subtle and manipulation-specific behavioural alternations[3,37]. Furthermore, it is also important to note here that all genetically modified *T. gondii* lines, including that of the recombinant wildtype *Tg*TH0, induced smaller intensities of behavioural alterations relative to true wild-type *T. gondii*, potentially implying that not only may the gene-modifying processes affect the viability of these recombinant lines, but that less subtle behavioural assays and/or statistical analyses may erroneously overlook the role of *Tg*TH. Indeed, the unique inclusion of a pure wildtype control group in our assays may be an important further explanation for the differences between studies, where previous research has proposed positive wildtype controls should always be included to provide a more comprehensive interpretation in *Tg*TH studies[49].

Nevertheless, there is still much to understand of the precise role of *Tg*TH in *T. gondii*'s mechanism(s) of action. Indeed, another study identified that *AAH* genes play an important role in parasite development during the sexual cycle, notably that of oocyst development, in the intestinal epithelium of the cat, rather than necessarily cyst formation[50]. Likewise, Afonso and colleagues found that *AaaH2 TH* deletion did not prevent cyst formation, indicating that this enzyme is not essential, either in sole or part, for in vivo cyst differentiation[34]. However, in addition to this being a single copy of duplicated genes, it is also important to note that some other studies have found avirulent *T. gondii* still induces behavioural changes after cyst clearance[9]. As Wang and colleagues also acknowledge, there are likely several pathways involved in *T. gondii* manipulation of host behaviour[20,33,51]. Indeed, there is mounting evidence that chronic infection impacts multiple physiological dimensions within its host, which directly or indirectly impact the regulation of neurotransmitters, including dopamine and beyond, in particular those of serotonin, GABA, glutamate and norepinephrine[20,47,52–54]. It seems likely then that the host behavioural phenotypes observed in *T. gondii* chronic infections are the net products of a complex interplay between distinct processes, of which dopamine dysregulation may only be one.

Our study may have been improved by using methods such as high-performance liquid chromatography to measure the dopamine level in the rats' brains in vivo, in complement to our in vitro here. However, we are confident of the biological applicability of our findings since previous in vitro and in vivo findings have found an association between *Tg*TH in rodent host brains[25], including, for example, the dose-dependent catalytic activity of *Tg*TH in dopamine synthesis[32] consistent with the OE dose-dependent behavioural changes observed in our study. Furthermore, dopamine neurotransmission has multiple facets where, for example, altered host dopamine levels can be attributed to increased turnover rate rather than increased concentrations[36]. Indeed, such increased turnover has also been used to explain why researchers need not detect transient dopamine changes, whilst behavioural alterations may still be apparent[33]. Nevertheless, future studies would benefit from this inclusion and should be performed using the same behavioural assays as we used here and with the more robust rat model, ideally also incorporating either *Tg*TH knockout and OE lines with higher expression relative to the wildtype, or ideally, given the observed viability of recombinant lines, comparisons across different avirulent *T. gondii* wildtype strains with inherently different *Tg*TH expression levels.

To conclude, using biologically appropriate assays and analyses, our finding here of a dose-dependent *Tg*TH overexpression increase in parasite-altered host behaviour provide unique support for the hypothesis of a contributory, if not sufficient, mechanistic role of *Tg*TH in *T. gondii*-associated manipulation. This, if confirmed, could have major implications for our mechanistic understanding of parasite manipulation of vertebrate host behaviour in general, with global applications across all infected hosts of this global multi-host parasite.

## Methods

### Ethical approvals and licences

The choice of host, parasite and behavioural assays used were designed to maximise accuracy, specificity and welfare[3,37]. All procedures were registered and approved by Imperial College's Animal Ethical and Welfare Review Board, classified as mild and performed under United Kingdom Home Office personal (MK & JPW) and project licences (JPW PPL 70/6761), in accordance with the Animal (Scientific Procedures) Act, 1986. All genetic modification procedures were registered and performed under GM licences from both Leeds University and Imperial College (GM & JPW). No animals showed any sign of clinical illness throughout the entire study.

### Host lines and maintenance

Rats were chosen as the model host rather than mice, as their increased level of resistance to *T. gondii* means that they generally display chronic, latent infection, rather than the more acutely virulent symptoms displayed in infected mice, and hence are believed to be more representative of human latent *T. gondii* infection[3,20,42]. The lister-hooded strain of rat was chosen because of its documented behavioural similarities to wild rats, especially in relation to neophobia traits[55–57]. Furthermore, their natural black on white coat markings aided both in the within-group visual identification of individuals, thereby avoiding the need for invasive tagging techniques which could impact rat's behaviour, and facilitated sensitive automated non-invasive monitoring of individual movement during behavioural trials. Both male and female rats were used for unconditioned (i.e., without previous learning) behavioural tests. Although most rodent studies in general tend to use single sexes, sex-difference in behavioural changes related to *T. gondii* infection has been reported in both humans and rodents[10,45,46,58]. Furthermore, studies have indicated that whilst laboratory male and female rodents respond similarly to fear-related experiments, females exhibits higher locomotor activity than male[41]. Hence, incorporating both sexes in *T. gondii* studies may be beneficial, as the subtle changes in parasite-related behaviours could be more apparent.

Sample size was calculated by Mead's Resource Equation E = N-T[59], where E is the number of error degrees of freedom, N the total number of degree of freedom, and T the number of treatments (please also see "supplementary note" within the supplementary information). A total of 85 rats (41 male and 44 female) were randomly selected and assessed across three laboratory rounds (see Table S1), with an even distribution of sexes among experimental lines (see Table S2).

Rats were obtained at three weeks of age from Harlan UK Ltd., and to minimize any generalized non-specific stress/anxiety level, were accustomed to handling from arrival. Rats were housed in groups of four, until male rats reached 500 + g in weight, after which males were housed in pairs. Female rats did not reach this weight. All rats were housed in large aerosol-controlled plastic cages with solid bottoms, with wood shavings and woodchips used as bedding. Bedding was changed two or three times weekly, and the cage sterilized weekly. Rats were fed and watered *ad libitum* on a pelleted rodent diet and sipper tubes. Environmental enrichment was provided through addition of items such as cardboard tunnels, wooden chew-sticks and shredded paper. Rats were infected at approximately three months of age.

### Parasite lines

The avirulent cyst-forming Pru (Prugniaud type II) strain of *T. gondii* was chosen as a model for latent toxoplasmosis in animals and humans −given that the acute pathology of virulent *T. gondii* strains would preclude observations of subtle behavioural change. Cyst-forming strains replicate the ecological scenario that evolutionary theory predicts benefit *T. gondii* transmission via intermediate hosts (parasite manipulation), as well as reflecting the behavioural changes predicted across humans and other secondary hosts (parasite constraint), given cyst persistence within the host CNS and potential for chronic low-grade inflammation and/or neuromodulation[3,51].

The parasites were maintained in culture as tachyzoites infecting human foreskin fibroblasts (Hs27 ATCC CRL-1634). These were maintained in Dulbeccos Modified Eagles Medium (DMEM) with 10% foetal bovine serum and 1% penicillin/streptomycin, at 37 °C and 5% $CO_2$, passaged every 5−7 days as required. Recombinant *Tg*TH lines of *T. gondii* overexpressing *TgAaaH2* were developed by transfection of Prugniaud Δhgxprt ΔKU80, a recombinant strain deficient in non-homologous recombination commonly used for genetic manipulation of *T. gondii*, with a plasmid containing TgAaaH2 flanked by the *Toxoplasma* tubulin gene promoter (a strong promoter) and the 3′ DHFR UTR, HGXPRT and GFP, by electroporation[60]. Restriction enzyme-mediated integration with plasmid linearised with Sac1 was used for integrating DNA into the genome. Recombinant parasites were selected with mycophenolic acid (25 mg/ml stock in ethanol) and xanthine (50 mg/ml stock in 0.1 M KOH). Liberated tachyzoites were sorted by FACS (BD Facsaria) for those with integrated plasmid and clonal parasites propagated and characterised.

Levels of recombinant *Tg*TH mRNA were quantified by quantitative RT-PCR of mRNA (relative to the parental Pru strain). Dopamine levels were then quantified in vitro via infected and uninfected rat PC12 cells (ATCC CRL-1721). Two replicate overexpression (OE) lines were uniquely selected here to represent contrasting levels of *Tg*TH expression, where: *Tg*TH-HIGH displayed the highest level of *Tg*TH and *Tg*TH-MID expressed a mid-range between the recombinant and parental lines. Two "wildtype" lines were selected as positive infection controls: one true wildtype (WT), serving as a wildtype positive control comparable to those used across a range of published studies[4,8], and one recombinant "wildtype" (deltaH ku80 PRU, hereafter referred to as *Tg*TH-0), which was a genetically modified laboratory strain, as other OE line, with multiple mutations, but excluding that of the *TgAaaH2* modification and thus served as an additional recombinant control in terms of evaluating any subtle impact of genetic modification, independent of *Tg*TH overexpression (Fig. 6a, b and Table 1).

(a)

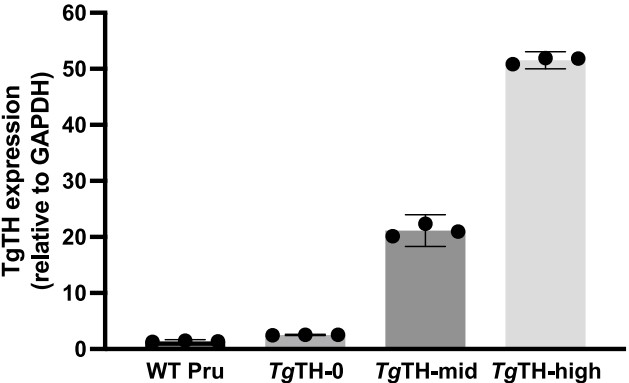

(b)

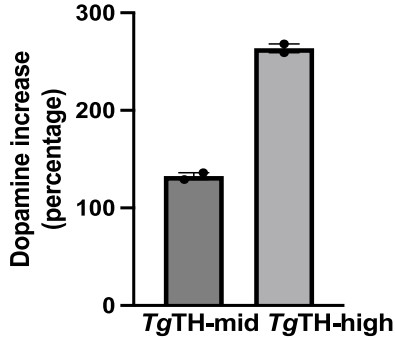

**Fig. 6 | Lines (strains) of Prugniaud type II (PRU) *Toxoplasma gondii* used.** **a** *Tg*TH gene expression (with SEMS) of WT Pru (wildtype), *Tg*TH-0 (delta H ku80 PRU recombinant wildtype), *Tg*TH-mid (overexpression strain), and *Tg*TH-high (overexpression strain), by RT-qPCR for TgAAAH2 quantified from a standard curve and relative to the housekeeping gene glyceraldehyde 3-phosphate dehydrogenase (GAPDH); ANOVA $p = 5.25 \times 10^{-13}$. **b** Increase in dopamine content (as a percentage) in neuronal cells infected with the mid- and high-*Tg*TH overexpression strains relative to parental *T. gondii* strain for generation of these strains with mean (SEM) of biological repeats.

Infection was conducted via intraperitoneal (IP) injection with 0.2 ml per rat containing approximately $1 \times 10^6$ tachyzoites in sterile phosphate-buffered saline (PBS). This volume was chosen based on a previous study, which also used IP inoculation of Pru strain tachyzoites in rats, and did not result in any morbidity, mortality, or complications[8]. Uninfected sham control rats were IP injected with 0.2 ml of sterile PBS.

After the completion of all behavioural studies, all experimental rats were euthanised by rising concentration of carbon dioxide and cervical dislocation. *T. gondii* antibodies were determined by the IgG indirect latex agglutination test (Toxoreagent; Eiken) and IgM direct agglutination test (BioMerieux, UK), where titres >1:32 were considered positive[40,61]. All rats exposed to *T. gondii* seroconverted, whilst those rats unexposed to *T. gondii* remained sero-negative. Likewise, chronic infection was confirmed by microscopic examination of tissue cysts in brain samples post-mortem from all *T. gondii* exposed rats, whilst no cysts were observed from unexposed control rat's brains.

### Behavioural assays: four-choice odour "Fatal Feline Attraction" test

The experimental setting was based on the standardized four-choice "Fatal Feline Attraction" assay[4,8,28]. Continuous exploratory behaviours

of rats were observed in 1 m × 1 m pens. 15 drops of water (neutral smell), cat urine (cat zone), rabbit urine (control for mammalian non-predator), and 0.5 × g of the rat's own urine-soiled bedding were deposited in woodchips within plastic nest boxes located in the respective four corners. To avoid positional biases, the four scented boxes were placed in different positions for each test. Trigene was used to clean and remove any olfactory cues from the used apparatus in between each test.

The experiment was conducted at 6 weeks post-inoculation, when the parasite has reached chronic infection, and hence the stage where subtle behavioural alterations first become established[62]. Due to logistical (space) constraints, the behavioural assays were performed across 3 experiment rounds. Rats were tested individually over 2.5 h, assessing a rat's response over time rather than just its initial response. Whilst the original "Fatal Feline Attraction" assays were performed over 10 h[4], subsequent research indicated 2.5 h to be the minimum duration for *T. gondii*-associated behavioural alterations to be reliably detected[28].

To prevent a human observer's presence from interfering with the results, behavioural tracking cameras were positioned by autopoles to the ceiling of the testing room, and recording commenced at a 5-min set point after the experimenter left. To ensure blinding, in addition to enabling more sensitive behavioural trait quantification (such as velocity), automated tracking hardware and software, Ethovision XT (Noldus, Wageningen, Netherlands), was used to record and analyse rats' behaviour. Each rat was provided with an anonymised ID, and thus the software processed the files in blind-coded form using the same algorithm for all rats and experimental lines. The centre pigmented spot of each rat was used to identify individuals and to detect the movements. Variables recorded included duration, distance, and frequency entering zones, as well as velocity of movement.

## Statistical analyses

Statistical analyses were carried out with R 4.4.3[63] in RStudio 2023.06.1 + 524[64]. Simple and multiple regression analyses were conducted within a Bayesian framework with the R packages brms 2.22.0[65–67] and cmdstanr 0.9.0[68], adopting the default brms priors. The Student $t$ distribution was used to model continuous variables (i.e., velocity in the cat zone, differences in velocity between cat zone and non-cat zone, and distance moved in the cat zone), the beta distribution was used to model proportions (i.e., proportion of time spent in the cat zone)[69], and the negative binomial distribution was adopted to model the count variable (i.e., frequency of entering the cat zone). In order to assess whether the experimental rounds had had an impact on the results from Control and WT (see Table S1), null models and models that included round as the only covariate were compared using leave-one-out cross-validation (LOO-CV) and Pareto-smoothed importance sampling (PSIS)[70,71] using the loo 2.8.0 package[72]. Models with "ELPD" differences (elpd_diff) below 4 were considered to have similar predictive power and the simplest of these models was selected. This approach for model comparison was also employed to assess whether line, sex and the interaction of both factors was associated with each of the outcomes. To this extent, five models (i.e., null, sex only, line only, line+sex, line*sex) were fitted for each outcome and compared with LOO-CV and PSIS as described above. Predictions were generated with the epred_draws command from the tidybayes v.3.0.7 R package[73], contrasts were computed between draws and the results were summarised (means and 95% credible intervals) with the mean_qi command using the ggdist 3.3.3 R package[74,75]. Additional R packages used include tidyverse 2.0.0[76] and bayesplot 1.13.0[77]. Graphical outputs were generated with the R packages ggplot2[78], ggdist 3.3.3[74,75] and RColorBrewer 1.1-3[79].

Three observations had extreme values (see Fig. S1: Proportion of time in the cat zone, male *Tg*TH0; Fig. S3: Velocity in the cat zone, female Control; Fig. S7: Frequency entering the cat zone, male, WT)

and were excluded from the results presented in the main text but were included in the results presented in the Supplementary Information (Figs. S2, S4 and S8). The inclusion of extreme values had limited qualitative impact on the velocity in the cat zone and the frequency entering the cat zone with respect to line, although did substantially raise the mean proportion of time spent in the cat zone of the *Tg*TH0 line due to a single male rat spending > 95% (4 standard deviations from the mean) of time in the cat zone (Fig. S1).

## Reporting summary

Further information on research design is available in the Nature Portfolio Reporting Summary linked to this article.

## Data availability

Source data are provided with the paper in full and are available via GitHub git@github.com:bcz/Toxoplasmosis_w.git and https://doi.org/10.5281/zenodo.17289079. Source data are provided with this paper.

## Code availability

The code required to reproduce the analyses presented in this study are available via GitHub git@github.com:bcz/Toxoplasmosis_w.git.

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

## Acknowledgements

We are very grateful to the wonderful support and care by the St Mary's animal technicians, and to Drs Knuepfer and Berdoy for comments on the text. This research was supported by the Stanley Medical Research Institute (J.P.W., G.M.), the Medical Research Council (M.K., J.P.W., P.H.L.L.) and the Royal Veterinary College (B.C.-U., A.-C.C.).

## Author contributions

Conceived the study: J.P.W. & G.A.M. Developed OE lines: G.B. under the supervision of E.P. & G.A.M. Performed the behavioural assays: M.K. under the supervision of J.P.W. & P.H.L.L. Performed the statistical analyses: B.C.-U. Wrote the first draft: J.P.W., B.C.-U. & A.-C.C. All authors, i.e., J.P.W., B.C.-U., M.W., G.A.M., P.H.L.L., G.C.B., E.P., A.-C.C. and M.K. contributed to revisions and/or approved the manuscript.

## Competing interests

The authors declare no competing interests.
