## [Transparent Peer Review file · Nature Communications]

The role of parasite-produced dopamine in *Toxoplasma gondii*-altered host behaviour

Corresponding Author: Professor Joanne Webster

Version 0:

Reviewer comments:

Reviewer #1

(Remarks to the Author)
Review

The role of dopamine in behavioral changes associated with latent *Toxoplasma gondii* infection, particularly those changes that appear adaptive from the perspective of the parasite's life cycle, is an important question in both theoretical parasitology (e.g., research on the manipulation hypothesis) and clinical medicine. As demonstrated by several independent meta-analytical studies, *T. gondii* infection is significantly associated with schizophrenia (as well as OCD), and elevated dopamine – levels are known to play a key role in the etiology of these disorders. The genome of *T. gondii* contains two genes encoding enzymes critical for dopamine synthesis, and studies have documented high dopamine concentrations both in tissue cysts and the surrounding neural tissue. However, some studies using transgenic *T. gondii* strains have shown that behavioral changes and total brain dopamine concentration in infected rodents do not depend on the presence of these genes. This controversy has persisted in the literature for roughly a decade.

The present study offers a promising and largely successful attempt to resolve this controversy. The authors again used transgenic parasite strains but infected rats (a much better model animal for study of latent toxoplasmosis) instead of mice and assessed the effects of infection using well-established and sensitive behavioral assays. They also included both positive and negative controls. Their results indicate that the presence of additional copies of the relevant genes affects the manifestation of specific behavioral symptoms of toxoplasmosis, and that the intensity of these symptoms correlates positively with the level of gene expression in the respective parasite strain. The authors also provide a compelling explanation for why previous studies using transgenic strains yielded different and seemingly contradictory results. Overall, the findings are highly valuable and the study has the potential to significantly influence the direction of research in this field.

Major Comments:

1. The authors infer a dose-dependent (expression-dependent) effect based on only three data points, one of which is a negative control. Furthermore, their model does not include a random effect of parasite line together with a fixed effect of gene expression level – though this is understandable given the low number of strains. While the results are consistent with the hypothesis of a positive relationship between behavioral alterations and gene expression, the strength of support for this hypothesis remains moderate.
2. A notable limitation is that test sessions were not randomized across parasite strains within each testing round (see Table S1). Although experimenters were blinded to group identity during scoring (and videos were likely evaluated in randomized order), this does not rule out objective bias introduced during testing itself – for example, differences across test runs.
3. Another methodological shortcoming is that behavioral testing was not repeated in an independent round with a longer interval. The lack of study preregistration is also unfortunate, although this remains common (and regrettable) across 90% of the current literature.
4. It is not explicitly stated whether the model containing the line*sex interaction also included the main effects of line and sex (though I assume it did). The importance of this interaction is also not reported. Including both sexes might be a major strength of this study, making it important to clarify whether sex moderated the behavioral effects of infection.
5. The authors mention conducting a power analysis but do not report the results.

Minor Issues:

6. The somewhat divergent pattern presented in Figure 2 should be discussed in the Discussion.
7. The Introduction or Discussion should mention previous findings of elevated dopamine levels in infected humans, i.e. reduced novelty-seeking trait, and the link to schizophrenia.
8. The sentence: "Lister-hooded rats were chosen due to their behavioural similarities with wild rats, especially in relation to neophobia traits [...] and clinical applicability to human latent toxoplasmosis" is poorly phrased. Lister-hooded rats are

unlikely to have direct clinical applications; rather, findings from such models may in the future inform our understanding of latent toxoplasmosis in humans.

9. The term “fitness” is used several times to describe reduced viability of transgenic *T. gondii* strains. However, fitness refers to the number of genes transmitted to future generations, not to survival or behavioral manipulation capacity. “Viability” would be more appropriate here.

10. Page 16: “Figures 6a&b & Table 1” – typos.

11. Table 1: Two columns appear to display more or less the same information and are redundant.

12. Figure 6: The boxplot is unnecessarily confusing. Standard practice is to place categories on the horizontal axis and the target variable on the vertical axis. I recommend switching the axes.

13. The *Toxoplasma gondii* species name is sometimes not italicized – especially in the reference list but also elsewhere. This should be corrected for consistency.

14. In the supplementary tables, the caption states that “Differences whose 95% credible interval does not include the value of zero are considered relevant and are highlighted in grey.” However, such values are not actually highlighted.

15. Citation formatting is inconsistent. Even if the original article used title case (capitalizing each major word), the reference list should use sentence case formatting for all entries.

16. The citation for Wang et al. (2015) is incorrect. It should read:

Wang, Z. T., Harmon, S., Dubey, J. P., & Sibley, L. D. (2015). Reassessment of the role of aromatic amino acid hydroxylases and the effect of infection by *Toxoplasma gondii* on host dopamine. *Infection and Immunity*, 83, 1039–1047.

<https://doi.org/10.1128/IAI.02465-14>

17. Sentence: “Our findings thereby provide unique support for the hypothesis of, at least in part, some mechanistic role of TgTH in *T. gondii*-associated behavioural changes, and we discuss our findings in terms of their theoretical and applied implications.”

Consider splitting into two clearer sentences and ending with a full stop.

Conclusion:

Despite the above comments, I consider this a highly important study that substantially advances our understanding of the role of dopamine synthesis genes in *T. gondii*-induced behavioral changes. I expect it will help resolve a longstanding controversy in the field.

(Remarks on code availability)

Reviewer #2

(Remarks to the Author)

I enjoyed the manuscript by Calvo-Urbano and colleagues titled “*Toxoplasma gondii* altered host behaviour: elucidating the role of parasite-produced dopamine”. My comments below are not in any specific order.

This manuscript asks if rate-limiting enzymes coded within the *Toxoplasma* genome underlie loss of fear often observed in *Toxoplasma*-infected rats. The authors note that a similar question has been earlier asked in mice using a knockout transgenic parasite. These earlier experiments did not find evidence of any association between removing (one of the copies) of the relevant gene and infecting mice with these transgenic parasites to observe behavioural change. The manuscript then points out that mice are not the appropriate study model for such experiments due to robust sickness in this species upon the acute phase of the infection. I wish to note my agreement with this assertion. It is common to see weight loss and changes in coat in mice during early phases of infection, reminiscent of robust inflammation. These effects make it very difficult to delineate if the observed effects are caused by sickness behaviour or are indeed parasitic manipulation. The use of a relatively outbred rat line is a good strength of this manuscript that stands out. Moreover, the authors have used two different overexpression transgenic parasite lines that are sufficiently controlled by not only uninfected but by a ‘null’ transgenic. The manuscript presents with robust and well-designed methodology.

I do have a bit of concern about the use of “feline attraction” to describe the phenomenon. What the manuscript shows is a loss of aversion with no clear preference of the group mean for predator odour. Yes, there are individual animals that do show preference, yet that is expected if one were to assume a spread around the mean at a proportion of 0.5. Some animals will have a proportion of more than 0.5 and some below 0.5 for purely stochastic or uncontrolled factors if there was a distribution centred on 0.5. I suggest it is a stretch to call it “attraction” rather than a loss of fear.

The authors rightly note in the discussion that they do not measure actual dopamine levels in the study. While this is a limitation, it does not distract from the conclusions of the manuscript. In fact, there are multiple processes where the supply of rate-limiting enzyme by the parasite might not raise dopamine at all and still create a phenotype. For example, an increase in dopamine supply might cause an increase in its reuptake at the presynaptic terminals. On the flip side, higher dopamine might initiate internalisation of post-synaptic receptors such that even an observed increase in dopamine might not result in increased synaptic transmission. The point I am trying to make is that not measuring dopamine in this manuscript is not a major limitation. Dopamine neurotransmission has multiple facets which are out of the scope of this manuscript. The data presented here is sufficient to make a judgement about the hypothesis of the study.

Overall, I believe the manuscript is a worthy addition to the mechanistic literature around *Toxoplasma*-induced behavioural manipulation.

(Remarks on code availability)

I do not feel I am equipped to review the code.

Version 1:

Reviewer comments:

Reviewer #1

(Remarks to the Author)

The authors have adequately addressed all of my previous comments. I now see only four minor issues that could still be corrected at the proofreading stage:

In the first sentence of the abstract, there appears to be an incorrect word order: the phrase currently reads "Certain can parasites...", which should likely be "Certain parasites can...".

In the sentence "Dopamine receptor antagonists commonly used in the treatment of patients with schizophrenia, such as haloperidol, show both anti-*T. gondii* tachyzoite activity in vitro [27], and have even been demonstrated to prevent the development of such behavioural changes in *T. gondii*-infected rats [20,28]." the word "both" is unnecessary, or the sentence should be rephrased to make the two clauses grammatically parallel (e.g. "have been shown both to ... and to ...").

In the Methods section, part of the text is still underlined ("Rats were chosen as the model host rather than mice, as their increased level of resistance to *T. gondii* means that they generally display chronic, latent infection, rather than the more acutely virulent symptoms displayed in infected mice, and hence are believed to be more representative of human latent *T. gondii* infection").

In the References, I noticed at least one occurrence of *Toxoplasma* written without italics.

In addition, a potentially more serious issue is that the link to the data and script on GitHub did not work for me and should be checked.

(Remarks on code availability)

The link to the data and script on GitHub did not work for me and should be checked.

REVIEWER COMMENTS

Reviewer #1 (Remarks to the Author):

Review

The role of dopamine in behavioral changes associated with latent *Toxoplasma gondii* infection, particularly those changes that appear adaptive from the perspective of the parasite's life cycle, is an important question in both theoretical parasitology (e.g., research on the manipulation hypothesis) and clinical medicine. As demonstrated by several independent meta-analytical studies, *T. gondii* infection is significantly associated with schizophrenia (as well as OCD), and elevated dopamine – levels are known to play a key role in the etiology of these disorders. The genome of *T. gondii* contains two genes encoding enzymes critical for dopamine synthesis, and studies have documented high dopamine concentrations both in tissue cysts and the surrounding neural tissue. However, some studies using transgenic *T. gondii* strains have shown that behavioral changes and total brain dopamine concentration in infected rodents do not depend on the presence of these genes. This controversy has persisted in the literature for roughly a decade.

The present study offers a promising and largely successful attempt to resolve this controversy. The authors again used transgenic parasite strains but infected rats (a much better model animal for study of latent toxoplasmosis) instead of mice and assessed the effects of infection using well-established and sensitive behavioral assays. They also included both positive and negative controls. Their results indicate that the presence of additional copies of the relevant genes affects the manifestation of specific behavioral symptoms of toxoplasmosis, and that the intensity of these symptoms correlates positively with the level of gene expression in the respective parasite strain. The authors also provide a compelling explanation for why previous studies using transgenic strains yielded different and seemingly contradictory results. Overall, the findings are highly valuable and the study has the potential to significantly influence the direction of research in this field.

Thank you very much.

Major Comments:

1. The authors infer a dose-dependent (expression-dependent) effect based on only three data points, one of which is a negative control. Furthermore, their model does not include a random effect of parasite line together with a fixed effect of gene expression level – though this is understandable given the low number of strains. While the results are consistent with the hypothesis of a positive relationship between behavioral alterations and gene expression, the strength of support for this hypothesis remains moderate.

Agreed, the number of strains used precluded this and hence we did not incorporate random effects - accordingly we state the moderate effects here.

2. A notable limitation is that test sessions were not randomized across parasite strains within each testing round (see Table S1). Although experimenters were blinded to group identity during scoring (and videos were likely evaluated in randomized order), this does not rule out objective bias introduced during testing itself – for example, differences across test runs.

Agreed, due to space/logistic reasons, we were unable to run all lines simultaneously, and were instead restricted to randomized lines versus controls and WT across three rounds. However, firstly,

in order to assess whether the experimental rounds had had an impact on the results from Control and WT (see Table S1), null models and models that included round as the only covariate were compared in our biostatistical analyses using leave-one-out cross-validation (LOO-CV) and Pareto-smoothed importance sampling (PSIS) using the `loo 2.8.0` package for R. Models with “ELPD” differences (`elpd_diff`) below 4 were considered to have similar predictive power and the simplest of these models was selected. In all cases the models without round were selected, suggesting that round did not have a significant effect on the outcomes.

Secondly, in an effort to explicitly ensure blinding, we did not use standard videos here but instead that of automated tracking hardware and software (Ethovision XT (Noldus, Wageningen, Netherlands) to record and analyse rats’ behaviour. Each rat was provided with an anonymised ID, and thus the software processed the files in blind-coded form using the same algorithm for all rats and experimental lines. Thus, in addition to enabling more sensitive behavioural trait quantification (such as velocity), such automated tracking software provided fully blinded data outputs for each individual rat over time, independent of experimental line.

This has been further clarified within our revised text.

3. Another methodological shortcoming is that behavioral testing was not repeated in an independent round with a longer interval.

I’m afraid we are not entirely sure what this referee is suggesting here. We assume perhaps that we could have re-run the four chamber ‘Fatal Feline Attraction’ assay after an interval. We have indeed performed such repeat assays in our previous trials (Webster et al., 2006), and whilst this also raises the issue of a potential generalized learning effect, which could differ between line, our past research using a repeat assay across a sub-group of rats has demonstrated that the impact of *T. gondii* on an individual rat’s innate response to feline urine is sustained over time. We have clarified this within our revised text.

The lack of study preregistration is also unfortunate, although this remains common (and regrettable) across 90% of the current literature.

Whilst formalized pre-registration was not required here (which we agree is an excellent concept and should be included in all future studies), all Home Office project and personal licences, together with GM board approvals, were approved before the trial commenced and the study was registered with IC Animal Ethics Review Board. This has been clarified in our revised text.

4. It is not explicitly stated whether the model containing the line*sex interaction also included the main effects of line and sex (though I assume it did). The importance of this interaction is also not reported. Including both sexes might be a major strength of this study, making it important to clarify whether sex moderated the behavioral effects of infection.

We did explicitly document the sex effect here, showing stronger effect in the more active females relative to males (e.g. Figure 4, Tables S9&S10). The models that included the interaction also included the main effects, line and sex. The models with the interaction were not significantly different to the simpler models that did not include the interaction term.

5. The authors mention conducting a power analysis but do not report the results.

Agreed and now incorporated.

Minor Issues:

6. The somewhat divergent pattern presented in Figure 2 should be discussed in the Discussion.

Agreed - although the findings were not actually divergent, given that the slightly higher intermediate value of the TgTH0 was not significant, this has been further expanded upon within our revised text.

7. The Introduction or Discussion should mention previous findings of elevated dopamine levels in infected humans, i.e. reduced novelty-seeking trait, and the link to schizophrenia.

Agreed and we have expanded further on this in our revised text.

8. The sentence: “Lister-hooded rats were chosen due to their behavioural similarities with wild rats, especially in relation to neophobia traits [...] and clinical applicability to human latent toxoplasmosis” is poorly phrased. Lister-hooded rats are unlikely to have direct clinical applications; rather, findings from such models may in the future inform our understanding of latent toxoplasmosis in humans.

Agreed and rephrased accordingly.

9. The term “fitness” is used several times to describe reduced viability of transgenic *T. gondii* strains. However, fitness refers to the number of genes transmitted to future generations, not to survival or behavioral manipulation capacity. “Viability” would be more appropriate here.

Agreed and amended in relation to the transgenic parasite lines and their apparent reduced ability to induce behavioural changes in their hosts here relative to wildtype parasite lines.

However, we also cite examples from the literature of clear reductions in fitness amongst other transgenic host and parasite lines relative to wildtype here. Furthermore, as we are using this behavioural assay as a proxy for parasite transmission to the next host stage, we do feel that fitness can also be cautiously used as an appropriate term, on occasion, here.

We have rephrased our revised text accordingly.

10. Page 16: “Figures 6a&b & Table 1” – typos.

Amended

11. Table 1: Two columns appear to display more or less the same information and are redundant.

Agreed, but we feel this format does most clearly reflect the differences between our parasite lines used, and thus would like to keep unless the editor feels strongly otherwise.

12. Figure 6: The boxplot is unnecessarily confusing. Standard practice is to place categories on the horizontal axis and the target variable on the vertical axis. I recommend switching the axes.

Agreed and amended.

13. The *Toxoplasma gondii* species name is sometimes not italicized – especially in the reference list but also elsewhere. This should be corrected for consistency.

Agreed and amended – notably within our references section (Endnote file).

14. In the supplementary tables, the caption states that “Differences whose 95% credible interval does not include the value of zero are considered relevant and are highlighted in grey.” However, such values are not actually highlighted.

The rows are indeed shown in grey (see Figures S4, S6, S8, S10 and S12). Perhaps there may be some confusion here in that the whole row is highlighted in grey, not just the differences, and hence we have rephrased for clarity.

15. Citation formatting is inconsistent. Even if the original article used title case (capitalizing each major word), the reference list should use sentence case formatting for all entries.

Agreed, as for point 13 above, we have amended our reference section and associated Endnote file accordingly

16. The citation for Wang et al. (2015) is incorrect. It should read:
Wang, Z. T., Harmon, S., Dubey, J. P., & Sibley, L. D. (2015). Reassessment of the role of aromatic amino acid hydroxylases and the effect of infection by *Toxoplasma gondii* on host dopamine. *Infection and Immunity*, 83, 1039–1047. <https://doi.org/10.1128/IAI.02465-14>

Apologies and amended accordingly.

17. Sentence: “Our findings thereby provide unique support for the hypothesis of, at least in part, some mechanistic role of TgTH in *T. gondii*-associated behavioural changes, and we discuss our findings in terms of their theoretical and applied implications.”
Consider splitting into two clearer sentences and ending with a full stop.

Agreed and this sentence has been rephrased/restructured accordingly.

Conclusion:

Despite the above comments, I consider this a highly important study that substantially advances our understanding of the role of dopamine synthesis genes in *T. gondii*-induced behavioral changes. I expect it will help resolve a longstanding controversy in the field.

Thank you very much.

Reviewer #2 (Remarks to the Author):

I enjoyed the manuscript by Calvo-Urbano and colleagues titled “*Toxoplasma gondii* altered host behaviour: elucidating the role of parasite-produced dopamine”.

Thank you very much.

My comments below are not in any specific order.

This manuscript asks if rate-limiting enzymes coded within the *Toxoplasma* genome underlie loss of fear often observed in *Toxoplasma*-infected rats. The authors note that a similar question has been earlier asked in mice using a knockout transgenic parasite. These earlier experiments did not find evidence of any association between removing (one of the copies) of the relevant gene and infecting mice with these transgenic parasites to observe behavioural change. The manuscript then points out that mice are not the appropriate study model for such experiments due to robust sickness in this species upon the acute phase of the infection. I wish to note my agreement with this assertion. It is common to see weight loss and changes in coat in mice during early phases of infection, reminiscent of robust inflammation. These effects make it very difficult to delineate if the observed effects are caused by sickness behaviour or are indeed parasitic manipulation. The use of a relatively outbred rat line is a good strength of this manuscript that stands out. Moreover, the authors have used two different overexpression transgenic parasite lines that are sufficiently controlled by not only uninfected but by a ‘null’ transgenic. The manuscript presents with robust and well-designed methodology.

Thank you very much.

I do have a bit of concern about the use of “feline attraction” to describe the phenomenon. What the manuscript shows is a loss of aversion with no clear preference of the group mean for predator odour. Yes, there are individual animals that do show preference, yet that is expected if one were to assume a spread around the mean at a proportion of 0.5. Some animals will have a proportion of more than 0.5 and some below 0.5 for purely stochastic or uncontrolled factors if there was a distribution centred on 0.5. I suggest it is a stretch to call it “attraction” rather than a loss of fear.

Here we were referring to the specific four-choice odour test, otherwise known as the ‘Fatal Feline Attraction assay’, which is now the standard replicable assay for *T. gondii*-altered host behaviour across species (from rodents to humans). Indeed, it was the unexpected finding of the original Berdoy et al., 2000 study that identified that infected rats displayed not simply a reduction in their innate aversion to the scent of felines, but an overriding attraction, displayed as an increased time spent within cat-odour areas relative to other neutral or non-definitive-host treated areas. Thus, whilst we accept that the current study was not explicitly measuring aversion verses attraction, we have further clarified within our revised text that the ‘Fatal Feline Attraction Assay’ refers to the behavioural assay chosen and why, and what our findings show here in relation to differences in time spent in cat-odour areas (relative to rabbit, water or own scented areas) etc.

The authors rightly note in the discussion that they do not measure actual dopamine levels in the study. While this is a limitation, it does not distract from the conclusions of the manuscript. In fact, there are multiple processes where the supply of rate-limiting enzyme by the parasite might not raise dopamine at all and still create a phenotype. For example, an increase in dopamine supply might cause an increase in its reuptake at the presynaptic terminals. On the flip side, higher dopamine might initiate internalisation of post-synaptic receptors such that even an observed increase in dopamine might not result in increased synaptic transmission. The point I am trying to make is that not measuring dopamine in this manuscript is not a major limitation. Dopamine neurotransmission has multiple facets which are out of the scope of this manuscript. The data presented here is sufficient to make a judgement about the hypothesis of the study.

Thank you very much and agreed, we have also further expanded upon this in our revised text.

Overall, I believe the manuscript is a worthy addition to the mechanistic literature around Toxoplasma-induced behavioural manipulation.

Thank you very much.

Reviewer #2 (Remarks on code availability):

I do not feel I am equipped to review the code.

Reviewer #1 (Remarks to the Author):

The authors have adequately addressed all of my previous comments.

Thank you.

I now see only four minor issues that could still be corrected at the proofreading stage:

In the first sentence of the abstract, there appears to be an incorrect word order: the phrase currently reads “Certain can parasites...”, which should likely be “Certain parasites can...”.

Agreed and amended.

In the sentence “Dopamine receptor antagonists commonly used in the treatment of patients with schizophrenia, such as haloperidol, show both anti-*T. gondii* tachyzoite activity in vitro [27], and have even been demonstrated to prevent the development of such behavioural changes in *T. gondii*-infected rats [20,28].” the word “both” is unnecessary, or the sentence should be rephrased to make the two clauses grammatically parallel (e.g. “have been shown both to ... and to ...”).

Agreed and amended.

In the Methods section, part of the text is still underlined (“Rats were chosen as the model host rather than mice, as their increased level of resistance to *T. gondii* means that they generally display chronic, latent infection, rather than the more acutely virulent symptoms displayed in infected mice, and hence are believed to be more representative of human latent *T. gondii* infection”).

Agreed and amended.

In the References, I noticed at least one occurrence of *Toxoplasma* written without italics.

Agreed and amended.

In addition, a potentially more serious issue is that the link to the data and script on GitHub did not work for me and should be checked.

Agreed and amended.

Reviewer #1 (Remarks on code availability):

The link to the data and script on GitHub did not work for me and should be checked.

Agreed and amended.